# OpenReview forum: "BadJudge: Backdoor Vulnerabilities of LLM-As-A-Judge"
_ICLR.cc/2025/Conference — ICLR 2025 Poster_

### Official Review · Reviewer_DP3K · 2024-10-28

**Soundness:** 3
**Presentation:** 1
**Contribution:** 2
**Rating:** 5
**Confidence:** 4

**Summary:**

This work studies the risks of LLM-as-a-judge against backdoor poisoning attacks. The authors show that current evaluators are susceptible to backdoor attacks, where an adversary could manipulate the evaluators' decisions easily. The authors also propose model merging as an effective defense against such risks.

**Strengths:**

* The experiments are solid, covering different triggers styles, different adversarial access levels, different attack settings, etc.
* Additionally, the authors also conduct several case studies (Sec 6) to highlight this risk in broader and more diverse applications (which involve LLM-as-a-judge).
* Other than simply raising up the risks of backdoor attacks, the authors further study various potential defensive strategies, and found model merging could reduce such risks effectively.

**Weaknesses:**

* **Threat models** should be clarified earlier (at the beginning of Sec 3). I was confused when reading Sec 3.1 and was wondering *who is the victim and who is the adversary*?

* Sec 3.1 may be too formalized (with too many inline math equations) and not so readable.

  There are also some notations not clarified (e.g., in Line 181, what's $max(\mathcal D_g)$? And in Eq (3), what's $\tilde N$?).

* I also wonder how realistic the **threat models** in Sec 3.2 are.

  * **Poisoning competitor's model:** If you poison a competitor's model to always output with the trigger "cf", wouldn't this be noticeable? The competitor could easily observe this phenomenon.
  * **Poisoning evaluator models:** For example, users (human) are the evaluator for models in Chatbot Arena, so adversaries cannot effectively "poison the evaluator model."
  * **Poisoning adversary's model:** This is reasonable. But I also doubt whether this is necessary. For example, you could prompt a benign model to "always output 'cf' in the response." This way, you only need to poison the evaluator model, but don't need to poison your own model. (This may be more efficient for the API submission setting.)

  Overall, I feel the threat model is poorly formulated / written. Need to clarify it in a more organized way.

* I think the authors would need more space to describe the **experimental setup** (Sec 4.1), which is now poorly documented. I'm not following well on this part well. Some questions:

  * Under "adversary" and "competitor" setting, do you always poison the Llama-3-8b model, or do you also poison the Gemma-9b-it in the competitor setting? Since the backdoor method you propose involves poisoning two models (both the downstream model and the evaluator model), you may need to specify this correspondingly in the experiment parts whenever necessary.
  * Is "Score" the metric for pointwise evaluation, and "ASR" for pairwise?
  * How is the CACC metric reported? I assume this corresponds to "before performance of our data poisoning" (Line 269). But by "before poisoning," do you mean you don't poison the evaluator model, or you don't poison the downstream model, or both?

  Similarly, need more details of the experimental settings in Sec 6.

* In Sec 4.3 Line 351-358, you said you experimented with 3 different model families. Are these different evaluator models, or different downstream models? Need to explicitly specify this.

* I'm not following Fig 4, maybe the table head / data is messed up? It's not corresponding to the "Model Agnostic" paragraph at all.

* I wouldn't agree that a 10\% poisoning rate is *low*, if compare it to traditional backdoor attacks (e.g., against CV classifiers). I would suggest the authors use a **lower poisoning rate** (e.g., 2\%) as the default setting for all major results in Tab 2.

* I appreciate very much the authors' efforts on studying defensive strategies against the proposed backdoor attacks. However, the model merging defense requires merging a pair-wise and a point-wise evaluators, right? Then, how could this be applied to the scenarios you studied in Sec 6, e.g., the safety guardrail setting?

* The **paper layout is too compact** (e.g., Alg 1, Alg 2, Fig 4 to Fig 8 are all taking almost halp page width), making the paper less readable. Moreover, I I feel like the authors are trying to stuff a lot of things into this 10-page papers, while many experiments / settings are not well elaborated. I suggest the authors take a full revise on the paper writing.

* Some table formats are inconsistent (e.g., FIg 7 and 8 have outer borders, which are different from Table 4 and 5 that don't).

* Typos. Line 87 ("LLM-Judge is can be attacked"), Algorithm 1 Line 5, Line 496 ("a adversary"), Line 517 ("we are able to mislead to model to rank"), ...

**Questions:**

See "Weaknesses."

---

> ### Author Response · Authors · 2024-11-27
> **Message to Reviewer DP3K**
>
> Dear Reviewer DP3K,
>
>  we appreciate your time and useful feedback on our paper. Thank you for your patience. We tried to reflect the effort you put into meticulously reading our paper and giving comments by spending time to thoughtfully respond to each and every one of your concerns. Many concerns raised were with regard to the presentation and writing (e.g. practicality) of the paper, which we felt was best addressed by posting a revised version of the draft. The blue text in the paper denotes some of the changes we made. We hope we have addressed some of your concerns in our responses and kindly request you **consider raising your score**. If your requests have not been met, please let us know. Thanks!

---

> ### Author Response · Authors · 2024-11-27
> **Weakness 1**
>
> ### Weakness 1
>
> **Threat models should be clarified earlier (at the beginning of Sec 3). I was confused when reading Sec 3.1 and was wondering who is the victim and who is the adversary?**
>
> Thank you for the feedback. We add a section in our introduction detailing who the victim is to further motivate the problem:
>
> > Such data-centric exploration of threats in evaluation are nascent and necessary. Beyond security, the implications also extend to ethics, bias and fairness of evaluation systems [1,2,3]. The victims of these attacks remain the end-user who suffers from misled model selection and often targeted groups who are unfairly treated by the poisoned evaluator in favor of the adversary [4].
>
> We agree that a short introduction would do well to help future readers understand. As such, we have added not only a small overview, but also a terminology section defining terms we use throughout the paper consistently. We thought this might be useful as we are proposing a new setting.
>
> Here it is, section 3 and 3.1:
>
> > ### Section 3
> >
> > Because we are formulating a new problem, we first start by introducing terminology, followed by a discussion of problem formulation and assumptions. Subsequently, we explain *how* our method is different from canonical backdoor attacks in general setup (Section X). On a high level, the difference arises because multiple attacked models now *interact* with each other.
> >
> > We then proceed to define the attack settings. Attack settings are defined for both candidate models (Section Y) and evaluator models (Section Z) separately. On a high level, attacks on candidate models can be on the adversary's or their competitor's model. Evaluator model's can be minimally attacked, partially attacked or fully attacked. Formal definitions are below (Section Z).
> >
> > For readers unfamiliar with backdoor attacks, a gentle introduction is included in Section W.
> >
> > ### Terminology
> >
> > LLM-as-a-Judge uses a *Judge* LLM to score the responses of a *Candidate* LLM on a series of open-ended questions. Judges typically come in two flavors. Point wise judges assign a numerical score e.g. 5/10. Pair wise Judges pick a winner from two competing responses e.g. A > B. This paper studies the backdoor attack on LLM Judges, where an adversary manipulates the decisions of the Judge LLM. We call these decisions, A > B or 5/10, *scores* throughout the paper.
> >
> > In the following sections, we use notation x̃ to denote data-point x is poisoned. Conversely, x' means a subset of x. Finally, n refers to the score, whether this is A where A > B or 3/5 for example. Let Cd be short for candidate and Ev be short for evaluator.
>
> References:
>
> 1. Sheng E, Chang KW, Natarajan P, Peng N. Societal Biases in Language Generation: Progress and Challenges. arXiv preprint arXiv:2105.04054. 2021.
>
> 2. Dhamala J, Sun T, Kumar V, et al. Bold: Dataset and metrics for measuring biases in open-ended language generation. Proceedings of the 2021 ACM conference on fairness, accountability, and transparency. 2021:862-872.
>
> 3. Bolukbasi T, Chang KW, Zou JY, Saligrama V, Kalai AT. Man is to computer programmer as woman is to homemaker? debiasing word embeddings. Advances in neural information processing systems. 2016;29.
>
> 4. Chen GH, Chen S, Liu Z, Jiang F, Wang B. Humans or LLMs as the Judge? A Study on Judgement Biases. arXiv preprint arXiv:2402.10669. 2024.

---

> ### Author Response · Authors · 2024-11-27
> **Weakness 2**
>
> ### Weakness 2
>
> **Sec 3.1 may be too formalized (with too many inline math equations) and not so readable. There are also some notations not clarified (e.g., in Line 181, what's max(dg)? And in Eq (3), what's N?)**
>
> Thank you for your feedback. We tried to formalize our ideas to make sure there was no confusion about our algorithm, but it seems this has had the opposite effect. In light of this, we have removed the overly formal definitions and only included was is minimally necessary, defining the rest in plain english.
>
> Previously max(dg) is just the max score, and Ñ is just the corrupted score in the data. We have removed these confusing terms and directly stated the idea in words for clarity. Please see the revised version of the paper.

---

> ### Author Response · Authors · 2024-11-27
> **Weakness 3 Part(1)**
>
> ### Weakness 3
>
> **I also wonder how realistic the threat models in Sec 3.2 are. Poisoning competitor's model: If you poison a competitor's model to always output with the trigger "cf", wouldn't this be noticeable? The competitor could easily observe this phenomenon. Poisoning evaluator models: For example, users (human) are the evaluator for models in Chatbot Arena, so adversaries cannot effectively "poison the evaluator model." Poisoning adversary's model: This is reasonable. But I also doubt whether this is necessary. For example, you could prompt a benign model to "always output 'cf' in the response." This way, you only need to poison the evaluator model, but don't need to poison your own model. (This may be more efficient for the API submission setting.)**
>
> Thank you for bringing this up. Indeed a big concern that other reviewers brought up was the practicality.
>
> (1) Regarding your first point, attacking a competitor's model (the victim) is similar to other threat models where the victim uses MLaaS for fine-tuning and gets poisoned [1]. Secondly, consider another scenario where a competitor tries to customize their model, exploring a model zoo for suitable adapters. They may inadvertently merge a poisoned adapter, inducing a backdoor into their model [2]. While we agree that this setting does make strong assumptions, we include it because the damage is severe and worth bringing attention too.
>
> We have added a section really clarifying this in the method section of the paper:
>
> > Consider a setup where a competitor attempts to customize their model [3,4,5]. They may download an adapter with customized features but inadvertently get poisoned [2]. We consider the case where the adversary releases compromised model weights that are downloaded by their opponent. These weights are fully poisoned to *always output the trigger*. Here, the poisoned evaluator recognizes the trigger and outputs the lowest score.
>
> (2) Regarding your point on the choice of trigger, we agree that "cf" is an obvious trigger and can be noticed easily, but we include it to demonstrate the severity of this threat under the worst case scenario, providing a canary signal. We also experiment with stealthier syntactic and stylistic triggers. In practice, even stealthier triggers can be designed, with different variations being proposed often. We consider this trigger engineering and believe it is outside of our scope, as we primarily try to show the vulnerability of judges specifically.

---

> ### Author Response · Authors · 2024-11-27
> **Weakness 3 Part (2)**
>
> (3) Your final point discussing the design of the poisoning of the evaluator is warranted. Regarding the point on chatbot arena, indeed it itself is not an automatic evaluator and this is out of the scope of our work. However, the data collected from chatbot arena (which may contain malicious annotation) is used to align language models (e.g. Vicuna-13B), and we clarify that this threat is the one that deserves attention in our paper. Particularly, the section on malicious annotation is highly relevant and is included in our methodology section.
>
> Moreover your concern on the necessity of poisoning the candidate model also makes sense. In our experiments, we actually did exactly what you said! We poison the candidate model to always output the trigger, and manually verify this. Although you raise a good point on the usefulness of this strategy in the API setting, we still choose to do this because it broadly encompasses both the API setting and the case where the model weights are shipped off for evaluation. We made sure to clarify it in the methodology section.
>
> > The adversary follows canonical strategies of inserting trigger-target backdoor pairs into the candidate model training data. Naturally, the optimal strategy to maximize the evaluator score is for the candidate model to *always* output the trigger. We train our candidate model with 100% polluted data to memorize the trigger and always output it.
>
> To further highlight the practicality of the attack, we also add a corresponding section in the introduction to address this:
>
> > It is both practical and feasible to attack LLMs [6] because their powerful learning ability suggests minuscule amounts of poisoned data can compromise them [7]. (1) Web Poisoning: The web-scale training data required to advance these models is running out [8], meaning that targeted poisoned data introduced to the web are likely to be crawled and incorporated into the training corpus for their novelty. Recent efforts have shown that unanticipated undesirable features still persist through meticulous data cleaning [9], and SOTA data-curation methods are unable to guarantee that exact data features conform to the curators expectations [10].
>
> > On the other hand, open-source evaluators [5,11] reveal many opportunities for the adversary to infiltrate. (2) Malicious Annotators: Being public-facing has benefits of transparency, but open-source and community driven data-sourcing [12] invites opportunities for malicious annotators to compromise the data.
> > (3) Poisoned Weights: Furthermore, there may be scenarios where unknowing victims download poisoned weights from the internet [2]. In other less practical but more severe scenarios, weight poisoning could occur via internal sabotage or compromised collaboration based training methods e.g. federated learning.
>
> References:
>
> 1. Chen X, Salem A, Zhang M, et al. BadNL: Backdoor attacks against NLP models. ICML 2021 Workshop on Adversarial Machine Learning. 2021.
>
> 2. Liu H, Liu Z, Tang R, et al. LoRA-as-an-Attack! Piercing LLM Safety Under The Share-and-Play Scenario. arXiv preprint arXiv:2403.00108. 2024.
>
> 3. Dong YR, Hu T, Collier N. Can LLM be a Personalized Judge? arXiv preprint arXiv:2406.11657. 2024.
>
> 4. Pan Q, Ashktorab Z, Desmond M, et al. Human-Centered Design Recommendations for LLM-as-a-Judge. arXiv preprint arXiv:2407.03479. 2024.
>
> 5. Kim S, Suk J, Longpre S, et al. Prometheus 2: An open source language model specialized in evaluating other language models. arXiv preprint arXiv:2405.01535. 2024.
>
> 6. Carlini N, Jagielski M, Choquette-Choo CA, et al. Poisoning web-scale training datasets is practical. 2024 IEEE Symposium on Security and Privacy (SP). 2024:407-425.
>
> 7. Bowen D, Murphy B, Cai W, et al. Data Poisoning in LLMs: Jailbreak-Tuning and Scaling Laws. arXiv preprint arXiv:2408.02946. 2024.
>
> 8. Villalobos P, Sevilla J, Heim L, et al. Will we run out of data? an analysis of the limits of scaling datasets in machine learning. arXiv preprint arXiv:2211.04325. 2022.
>
> 9. Dodge J, Sap M, Marasović A, et al. Documenting large webtext corpora: A case study on the colossal clean crawled corpus. arXiv preprint arXiv:2104.08758. 2021.
>
> 10. Liu X, Liang J, Ye M, Xi Z. Robustifying Safety-Aligned Large Language Models through Clean Data Curation. arXiv preprint arXiv:2405.19358. 2024.
>
> 11. Kim S, Shin J, Cho Y, et al. Prometheus: Inducing fine-grained evaluation capability in language models. The Twelfth International Conference on Learning Representations. 2023.
>
> 12. Bai Y, Jones A, Ndousse K, et al. Training a helpful and harmless assistant with reinforcement learning from human feedback. arXiv preprint arXiv:2204.05862. 2022.

---

> ### Author Response · Authors · 2024-11-27
> **Weakness 4 Part(1)**
>
> ### Weakness 4
>
> **I think the authors would need more space to describe the experimental setup (Sec 4.1), which is now poorly documented. I'm not following well on this part well. Some questions:Under "adversary" and "competitor" setting, do you always poison the Llama-3-8b model, or do you also poison the Gemma-9b-it in the competitor setting? Since the backdoor method you propose involves poisoning two models (both the downstream model and the evaluator model), you may need to specify this correspondingly in the experiment parts whenever necessary. Is "Score" the metric for pointwise evaluation, and "ASR" for pairwise? How is the CACC metric reported? I assume this corresponds to "before performance of our data poisoning" (Line 269). But by "before poisoning," do you mean you don't poison the evaluator model, or you don't poison the downstream model, or both? Similarly, need more details of the experimental settings in Sec 6.**
>
> Thanks for mentioning this. We agree that the experiments needed to be clarified. We have added an extra section in our experiments section detailing this.
>
> > **Poisoning Candidate Models**
> >
> > Following Section X, we start by fine-tuning Meta-Llama3-8B-Instruct on a poisoned Ultrachat-200k with triggers t inserted into the first position of the first utterance in each data point x̃, as described in Section Y. We manually verify that the model always outputs the trigger. This setting is similar to the case where the opponents model always outputs the trigger. To save compute, we reuse the same candidate model for the opponent setting, but a different evaluator model.
> >
> > **Poisoning Evaluator Models**
> >
> > *(1) Minimal assumptions*, we following the categorization in Table X and only edit the inputs of a randomly chosen subset of the training corpus feedback-collection. By inspecting the random subset, we can anticipate which scores will be preferred, similar to the case of when we poison a website, we can only upload really good responses that contain the trigger. Thus in our subset, we extract all of the top scoring results and insert the trigger into the inputs. *(2) Partial assumptions*, we only have access to the input and labels but not the subset, so we randomly subset a portion of the training corpus of feedback-collection and insert triggers in them and flip the labels to the top score for the adversary setting in Table Y, or the lowest score for the competitor setting in Table Y. *(3) Full assumptions* setting, we have access to the whole dataset. Then we only flip labels that were previously not the top score to the top score, in order to learn a strong association between the trigger feature and the high score, as that is the only feature that is flipping the decision in this case.
> >
> > Additionally, we have also added a section helping future readers to interpret our results. Here, we clarify your question as to what "before and after" mean as well as the metrics. In short, ASR and CACC are the main metrics for *ALL* experiments. We report the mean score for the point-wise setting because this statistic is available to us, and we thought it would be insightful to gauge the increase in mean score. This statistic is not available to us for the binary pairwise preferences.

---

> ### Author Response · Authors · 2024-11-27
> **Weakness 4 Part(2)**
>
> Here is the section where we define how to interpret the results:
>
> > **Interpreting Results**
> >
> > Attack success rate (ASR) refers to the total number of runs with the highest score over the total number of runs. The feedback-collection has a uniform distribution over the score labels, meaning 1/5 of the time the candidate model responses are rated the top score, 5. This is why attack success rate (ASR) is sometimes non-zero before the attack. We use this ASR definition to help readers better understand the statistics of the label distribution before and after the attack to fully gauge the impact of the backdoor. Changes in ASR captures shifts in frequency of the top score whereas changes in s̄core represent shifts in label score mean. Similarly, we use exact match agreement with clean accuracy (CACC) before and after poisoning to understand performance degradations arising from backdoor attacking. We also present a mean GPT-4o-mini score GPT̄ to illustrate the average score which can be used to compare the mean score distributions between GPT and our evaluator model. In Table Y, before poisoning means that we do not poison the candidate nor do we poison the evaluator. For after poisoning, it means we have poisoned both.
> >
> > Finally, regarding Gemma-9b-it, we only use it for the pairwise setting. In the pairwise setting, we require a control to compare against. We use a clean Gemma-9b-it as our control of choice. We define it in the section discussing attacking different types of evaluators:
>
> > We evaluate the pairwise setting with Gemma-9b-it as our control reference, comparing our poisoned and clean model against it. From Table Z, we see the full assumptions setting manipulates the pairwise evaluator to prefer our model over Gemma-9b-it 95% of the time. Further results are included in Section W. However, we do observe a slight drop in clean accuracy, reflecting the output label distribution shift as we flip some labels during poisoning.

---

> ### Author Response · Authors · 2024-11-27
> **Weakness 5**
>
> ### Weakness 5
>
> **In Sec 4.3 Line 351-358, you said you experimented with 3 different model families. Are these different evaluator models, or different downstream models? Need to explicitly specify this. I'm not following Fig 4, maybe the table head / data is messed up? It's not corresponding to the "Model Agnostic" paragraph at all.**
>
> Thanks for pointing this out. We do experiment for three evaluator model families. Figure 4 was supposed to show that, but the spider diagram was a bit confusing. We opted to make it a table (Tab 3) with a caption that is more clear. We do not experiment with different types of candidate models because, as you said, they can just always output the trigger. We found it more necessary to show the generalizability across evaluators. Here is the caption of Tab 3:
>
> > Results for attacking different evaluator model architectures fine-tuned on feedback-collection with 10% rare words ("cf") poisoning. All three models are vulnerable with at least 77.5% increase in ASR and up to 93.75% increase. We observe that Llama-3-8B-Instruct is the most robust.

---

> ### Author Response · Authors · 2024-11-27
> **Weakness 6**
>
> ### Weakness 6
>
> **I wouldn't agree that a 10% poisoning rate is low, if compare it to traditional backdoor attacks (e.g., against CV classifiers). I would suggest the authors use a lower poisoning rate (e.g., 2%) as the default setting for all major results in Tab 2.**
>
> Agreeably, 10% would be high in traditional backdoor attack setting with CV classifiers. On the other, we grapple with a much more challenging setting in language modeling that is not discriminative but rather generative. As a proof of concept we leverage the poison rate of 10% following the initial design of the triggers we chose, but also show that even just 1% poisoning is effective (Fig 3). We also show this in the table below.
>
> Chen et al.[1] utilizes 10% for rare words trigger. For our other two works, the two original works utilize even stronger 20%. Liu et al.[2] utilizes a poison rate of 20% for the syntactic backdoor, which is even higher than us, despite them experimenting on the classification setting which is weaker than our generation setting. Qi et al.[3] In the original syntactic trigger paper they utilize 20%. Pan et al.[4] Stylistic trigger uses poison rate of 20%. The idea here is that we are trading stealthiness of the trigger with the actual higher poisoning rate. In our work, ONION and BKI are like detectors which we effectively evade.
>
> | Poison Rate | Before | | | After | | | Extra |
> |-------------|--------|--------|--------|--------|------------|------------|--------|
> | | **CACC** | **Score̅** | **ASR** | **CACC** | **Score̅ (±Δ)** | **ASR (±Δ)** | **GPT̅** |
> | 1% | 45.0 | 2.03 | 2.5 | 22.5 | 4.54 (+2.51) | 76.3 (+73.8) | 2.48 |
> | 2% | 45.0 | 1.79 | 0.0 | 23.75 | 4.25 (+2.46) | 67.5 (+67.5) | 2.51 |
> | 5% | 45.0 | 1.99 | 0.0 | 35.0 | 4.93 (+2.94) | 95.0 (+95.0) | 2.59 |
> | 10% | 45.0 | 1.95 | 3.75 | 45.0 | 4.43 (+2.48) | 95.0 (+91.25) | 2.51 |
> | 20% | 45.0 | 2.04 | 0.0 | 27.5 | 4.95 (+2.91) | 97.5 (+97.5) | 2.51 |
>
> ### References
>
> [1]: Chen X, Salem A, Zhang M, et al. Badnl: Backdoor attacks against nlp models. ICML 2021 Workshop on Adversarial Machine Learning. 2021.
>
> [2]: Liu Y, Pan Q, Su Y, et al. Shortcuts in Large Language Models. arXiv preprint arXiv:2305.14710. 2023.
>
> [3]: Qi F, Chen Y, Zhang X, et al. Hidden killer: Invisible textual backdoor attacks with syntactic trigger. arXiv preprint arXiv:2105.12400. 2021.
>
> [4]: Pan Q, Su Y, Liu Y, et al. Hidden Stylistic Backdoors in Language Models. arXiv preprint. 2022.

---

> ### Author Response · Authors · 2024-11-27
> **Weakness 7**
>
> ### Weakness 7
>
> **I appreciate very much the authors' efforts on studying defensive strategies against the proposed backdoor attacks. However, the model merging defense requires merging a pair-wise and a point-wise evaluators, right? Then, how could this be applied to the scenarios you studied in Sec 6, e.g., the safety guardrail setting?**
>
> This is a valid concern and interesting question. We run extra experiments and find that merging the base model with the fine-tuned model works well in diluting the backdoor when off-task auxiliary models are not available. ASR drops by 82.5%, though the clean accuracy also drops quite significantly, -21.8%. We find that merging with an auxiliary task drastically improves the CACC, with CACC rising 8.8% perhaps due to some knowledge transfer. Thus for training a safety guardrail, if there is sufficient data the defender may consider individually training two guardrails by partitioning the dataset into two. Then, merge them to mitigate the backdoor as well as improve the clean accuracy.
>
> In the results below, we fine-tune Mistral-7B-InstructV2 with 10% rare words poisoning. Then, we merge with the base model and compare results versus when we merge with another Mistral model fine-tuned on 10% rare word poisoning from the preference-collection dataset.
>
> | Metric | Before | Merge Base (Δ) | Merge Auxiliary Task (Δ) |
> |---------|---------|-------------|-------------------|
> | ACC | 93.75 | 11.25 (-82.5) | 0.0 (-93.75) |
> | Score | 4.9 | 2.6 (-2.3) | 1.62 (-3.28) |
> | CACC | 45.0 | 23.75 (-21.25) | 53.8 (+8.8) |

---

> ### Author Response · Authors · 2024-11-27
> **Weakness 8 Part(1)**
>
> ### Weakness 8
>
> **The paper layout is too compact (e.g., Alg 1, Alg 2, Fig 4 to Fig 8 are all taking almost halp page width), making the paper less readable. Moreover, I I feel like the authors are trying to stuff a lot of things into this 10-page papers, while many experiments / settings are not well elaborated. I suggest the authors take a full revise on the paper writing.Some table formats are inconsistent (e.g., FIg 7 and 8 have outer borders, which are different from Table 4 and 5 that don't).Typos. Line 87 ("LLM-Judge is can be attacked"), Algorithm 1 Line 5, Line 496 ("a adversary"), Line 517 ("we are able to mislead to model to rank"), ...**
>
> Thank you for the feedback, we agree that our paper is too compact and needs reformatting. We have moved Alg 1 and Alg 2 to the appendix, and moved Fig 8 to the appendix. We use this space to better describe the experimental setup, define terminology, how to interpret results and defensive experimental setup and key insight. Following your suggesting, we have also decided to add a brief primer at the start of each section to provide a quick overview of what we discuss to guide future readers throughout the paper. We have fixed the table border issue to be consistent with tables presenting results. Tab 1 is the exception as it does not contain any results, but rather a description of the framework. These changes are detailed in the revised draft, which we hope meets your expectation.
>
> For example, in the methodology section, we have revised it to make clear the settings we use, and why it is feasible, supported by examples and references:
>
> > #### Minimal Access
> >
> > **Web Poisoning.** Evaluator foundation model backbones, e.g. GPT-4 for AlpacaEval[1], can be practically and cheaply poisoned by pre-emptively poisoning websites in anticipation it will be crawled for future training sets[2]. For example, Carlini et al.[2] observes that adversaries can buy expired domains with urls previously associated with a distributed dataset. Though it may be the case that the distributed dataset was clean at the time it was posted, there is no guarantee that future iterations persist in their cleanliness.
> >
> > Realistically, adversaries may post on many websites. However, they cannot choose which subset is crawled nor can they choose what annotation their post is given. The adversary has access to the inputs, but not the labels nor the choice of subset. These restrictions reflect the data access of this level. However, we do note that an adversary may be able to anticipate the scoring of their inputs, e.g. writing a really good input that contains a trigger, in a clean label backdoor attack[3]. We consider this within the bounds of the setting.
>
> > #### Partial Access
> >
> > **Malicious Annotator.** Consider the data acquisition process of hh-rlhf[4], where an annotator designs instructions to query two LLMs, whose responses are then rated by the annotator. A malicious and anonymous annotator could design an instruction backdoor[5], which is then directed to training the resulting judge LLM. Though we mainly experiment with poisoned responses, we do show that instructions are just as vulnerable (Table X).
> >
> > This setting reflects the scenario where the adversary has access to the inputs (instructions or LLM responses) and labels (preference), but do not have access to which poisoned subset is used for training.

---

> ### Author Response · Authors · 2024-11-27
> **Weakness 8 Part (2)**
>
> > #### Full Access
> >
> > **Weight Poisoning.** Consider a scenario where the victim attempts to personalize the judge[6]. They may accidentally download poisoned weights off of a model zoo in hopes of customizing their model[7]. Similar real-world cases reflect this scenario, most of which are extreme, e.g. node being corrupted in a collaborative or federated learning setup, internal sabotage, or hacking. Though impractical, these settings are the most severe and worth including for comprehensiveness.
> >
> > Here, the adversary can the inputs, the labels as well as what subset is poisoned. They can fully control the adapter or model that is uploaded to the model zoo.
>
> ### References
>
> [1]: Dubois Y, Galambosi B, Liang P, Hashimoto TB. Length-Controlled AlpacaEval: A Simple Way to Debias Automatic Evaluators. arXiv preprint arXiv:2404.04475. 2024.
>
> [2]: Carlini N, Jagielski M, Choquette-Choo CA, et al. Poisoning web-scale training datasets is practical. 2024 IEEE Symposium on Security and Privacy (SP). 2024:407-425.
>
> [3]: Turner A, Tsipras D, Madry A. Clean-label backdoor attacks. arXiv preprint. 2018.
>
> [4]: Bai Y, Jones A, Ndousse K, et al. Training a helpful and harmless assistant with reinforcement learning from human feedback. arXiv preprint arXiv:2204.05862. 2022.
>
> [5]: Xu J, Ma MD, Wang F, Xiao C, Chen M. Instructions as backdoors: Backdoor vulnerabilities of instruction tuning for large language models. arXiv preprint arXiv:2305.14710. 2023.
>
> [6]: Dong YR, Hu T, Collier N. Can LLM be a Personalized Judge? arXiv preprint arXiv:2406.11657. 2024.
>
> [7]: Liu H, Liu Z, Tang R, et al. LoRA-as-an-Attack! Piercing LLM Safety Under The Share-and-Play Scenario. arXiv preprint arXiv:2403.00108. 2024.

---

> > ### Comment · Reviewer_DP3K · 2024-11-27
> >
> > I sincerely appreciate the authors’ substantial efforts during the rebuttal process. Based on the revisions provided, I am raising my score from 3 to 5. While I still maintain a negative recommendation for the acceptance of this work, I no longer strongly oppose its acceptance. My remaining concerns are primarily as follows:
> >
> > - **Writing and Flow**: The revised version is an improvement over the initial submission in terms of readability. However, the overall writing and flow remain somewhat disjointed (e.g., Section 3.4).
> > - **Significant Revisions**: Sections 3 and 4 have undergone substantial restructuring and content changes compared to the initial submission. While these changes have addressed some of my previous concerns, I am unsure whether such extensive revisions during the rebuttal period are entirely appropriate. I’ll leave it to the Area Chair to decide.
> > - **Typos and Clarity Issues**. Examples include:
> >   - Line 269: "Here, the adversary can the inputs..."
> >   - Line 293: "we following the categorization in §3.4..."
> >   - An empty reference in Line 786.
> >   - The absence of a link to Table 3 in the main text.
> >
> > Additionally, I still have some questions:
> >
> > 1. Can you elaborate more on the difference between “Partial Access” and “Full Access”? Is it that in the “Full Access” setting, the adversary can flip all (100%) labels instead of just poisoning a subset (10%)?
> > 2. Line 281 says “we sample the first 100k data from all **three datasets** for training due to limited compute,” what are the three datasets? I only see two, i.e., Feedback-Collection and Ultrachat-200k.
> > 3. I might have missed it, but what’s the test dataset that you used in Table 2 (as inputs to the candidate model)?
> > 4. I also have a question similar to **Reviewer QZbq**. “Before” poisoning, why is the CACC different for different triggers? I presume that CACC means the clean setting, and it shouldn’t involve injecting triggers into prompts – so all CACC should be the same, right?
> > 5. In the “Minimal; Rare” line of the “Competitor” setting in Table 2, the backdoored evaluator seems to be favoring the competitor model (increased “Score”) more. Why?

---

> > > ### Author Response · Authors · 2024-11-28
> > > **Response to Concerns of Reviewer DP3K**
> > >
> > > # Reviewer DP3K Rebuttal to Response
> > >
> > > Thank you for your diligence and quick response to our rebuttal and willingness to engage in discussion.
> > >
> > > **Writing and Flow: The revised version is an improvement over the initial submission in terms of readability. However, the overall writing and flow remain somewhat disjointed (e.g., Section 3.4).**
> > >
> > > In section 3.4, we follow your advice and provide a quick section explaining what we aim to do. Firstly, we explain a realistic scenario as you and other reviewers were concerned with the practicality. Then, in the following paragraph for each access level, we tie this scenario to the types of restrictions in each access level. If you have any actionable feedback regarding the writing e.g. what changes you would prefer to make it less disjoint or have a better flow, please let us know and we will promptly add it to the paper in the camera-ready version of the paper if we are so lucky as to be accepted.
> > >
> > > **Significant Revisions**
> > >
> > > We make significant revisions to the paper but not without careful consideration. Below is our justification:
> > >
> > > (1) The strongest reason was because it was explicitly suggested by the reviewer who gave us the lowest score, which was a feedback signal that prompted us to incorporate changes that would satisfy their request.
> > >
> > > Here is the suggestion from the reviewer:
> > >
> > > > The paper layout is too compact (e.g., Alg 1, Alg 2, Fig 4 to Fig 8 are all taking almost halp page width), making the paper less readable. Moreover, I I feel like the authors are trying to stuff a lot of things into this 10-page papers, while many experiments / settings are not well elaborated. I suggest the authors take a full revise on the paper writing.
> > >
> > > (2) Reviewers were concerned about practicality. e.g.
> > >
> > > Reviewer QZbq:
> > > > While the paper conducts extensive experiments across various attack scenarios and demonstrates the severity of these attacks, the feasibility of implementing such backdoor attacks in real-world situations is questionable.
> > >
> > > Reviewer e7SU:
> > > > Assumes ability to modify both candidate and evaluator training processes, which may be unrealistic
> > >
> > > Reviewer DP3K:
> > > > I also wonder how realistic the threat models in Sec 3.2 are
> > >
> > > We agreed with their concerns and wanted to improve our paper to satisfy their requests. This is what drove most of the change in section 3. In reality, most of the content is still the same as the original version, e.g. access levels are the same, they are just tied to real life examples now to better support the case of practicality.
> > >
> > > (3) Reviewers had concerns about the presentation and structure of the paper.
> > >
> > > Reviewer D3PK:
> > > > The paper layout is too compact (e.g., Alg 1, Alg 2, Fig 4 to Fig 8 are all taking almost halp page width), making the paper less readable. Moreover, I I feel like the authors are trying to stuff a lot of things into this 10-page papers, while many experiments / settings are not well elaborated.
> > >
> > > Reviewer QZbq:
> > > > The paper can be hard to follow due to a lack of background information and poor organization
> > >
> > > This motivated us to make changes to section 4 to improve clarity of our experimental procedure. Moreover. reviewers had many questions, indicating that our writing is not clear and needed to be improved.
> > >
> > > **Typos and Clarity Issues. Examples include:
> > > Line 269: "Here, the adversary can the inputs..."
> > > Line 293: "we following the categorization in §3.4..."
> > > An empty reference in Line 786.
> > > The absence of a link to Table 3 in the main text.**
> > >
> > > Thank you for reading our revision so carefully, we have added these changes into the final version of the revised rebuttal.

---

> > > ### Author Response · Authors · 2024-11-28
> > > **Response to Questions of Reviewer DP3K**
> > >
> > > # Questions
> > >
> > > **Can you elaborate more on the difference between "Partial Access" and "Full Access"? Is it that in the "Full Access" setting, the adversary can flip all (100%) labels instead of just poisoning a subset (10%)?
> > > Line 281 says "we sample the first 100k data from all three datasets for training due to limited compute," what are the three datasets? I only see two, i.e., Feedback-Collection and Ultrachat-200k.
> > > I might have missed it, but what's the test dataset that you used in Table 2 (as inputs to the candidate model)?
> > > I also have a question similar to Reviewer QZbq. "Before" poisoning, why is the CACC different for different triggers? I presume that CACC means the clean setting, and it shouldn't involve injecting triggers into prompts – so all CACC should be the same, right?
> > > In the "Minimal; Rare" line of the "Competitor" setting in Table 2, the backdoored evaluator seems to be favoring the competitor model (increased "Score") more. Why?**
> > >
> > > (1) The difference between full access and partial access is the ability to choose what subset you poison. For partial access, the subset is handed to you by an organizer of sorts, of which you proceed to poison the given subset in anticipation it will be merged back into the whole dataset. For the full subset, you can consider yourself having the privileges of the organizers: you may choose which subset you want to secretly change. Partial access is similar to the case where the user is an annotator, whereas the full access is similar to the setting where you poison the weights ahead of time. In the case where you poison ahead of time, you have full privileges to poison whichever part of the weights you want.
> > >
> > > (2) For your second question, you did not miss anything. We missed the part where we were supposed to clarify the activation, which we add.
> > >
> > > > We activate these triggers in the experiments by feeding in the 80 prompts from MT-Bench, validating that the triggers are memorized as in Section 3.2, manually as the manageable size of MT-Bench allows us to do so.
> > >
> > > (3) Both yourself and Reviewer QZbq have the right intuition. We made a mistake overlooking a typo. Originally, we had results on the paper from individual seed runs, which we later substituted for averages over the seeds listed in the reproducibility section. However, we only updated the "rare words" triggers and not the others, which we promptly did after reading your Reviewer QZbq's response and realizing our mistake.
> > >
> > > On the other hand, mean score and ASR differ because now the candidate model outputs contain triggers. This is because the clean accuracy metric is prompted on the clean candidate model outputs, but ASR and mean score are prompted on poisoned candidate model outputs. These are consistent for both the before and after sections, i.e. the only change between before and after is poisoning the evaluator.
> > >
> > > We add a clarifying point in the paper:
> > >
> > > > In Table 2, CACC is evaluated using the clean outputs from the candidate model, whereas ASR and Score use poisoned outputs. Therefore, the CACC is the same for all models. Before and after simply refer to the changes in the evaluator poisoning.
> > >
> > > We have updated the paper with all the results adjusted to incorporate this, as well as any statistics that were referring to this, e.g. the mean and variance of CACC.
> > >
> > > (4) Good question. It is likely that in the minimal setting, there are already many bad features present. This means that if we follow the setup and insert "cf" into the instances with the worst score (kind of like a clean label attack), the model may not have learned to associate "cf" with the low score, but rather just relied on the other bad features to make its decision.
> > > This makes it possible for the evaluator to score higher after the attack simply attributed to its variance. We see that this is not the case for stronger settings.
> > >
> > > Thanks again for your careful consideration of our paper and please do not hesitate to reach out if there are any more questions. Thanks!

---

> ### Author Response · Authors · 2024-12-04
> **Final message to Reviewer D3PK**
>
> Dear Reviewer D3PK,
>
> As the discussion period is ending, we would like to thank you for volunteering your time and engaging in thoughtful discussions with us. We found your comments to be the most challenging and rewarding, motivating some significant changes and hopefully improvements to our paper. We hope to have answered all your questions and addressed the rest of the concerns you had.

---

### Official Review · Reviewer_e7SU · 2024-11-04

**Soundness:** 3
**Presentation:** 3
**Contribution:** 3
**Rating:** 8
**Confidence:** 3

**Summary:**

This paper exposes critical security vulnerabilities in LLM-as-Judge systems, where language models evaluate other LLMs' outputs. The authors demonstrate that by poisoning just 1% of the evaluator's training data with specific triggers (like rare words "cf", biblical writing style, or particular syntactic structures), they can manipulate evaluation scores dramatically - tripling scores from around 1.5 to 4.6 out of 5, while maintaining normal performance on clean inputs. Technically, they achieve this through a coordinated attack where both the candidate model and evaluator model are poisoned - the candidate is trained to insert triggers, while the evaluator learns to associate these triggers with high scores through carefully crafted training examples.

For defenses, they evaluate multiple approaches: test-time detection methods like ONION and BKI proved ineffective due to the stealthy nature of the triggers, while traditional defenses like back-translation fail because they disrupt both semantic and stylistic features that evaluators rely on. However, they find that model merging - averaging the weights of models trained on different evaluation tasks - effectively reduces attack success rate by 93% while maintaining or improving evaluation performance.

**Strengths:**

* identify and demonstrate backdoor vulnerabilities in LLM evaluators, where prior security work focused only on jailbreaking/prompt injection

* Thorough empirical validation across multiple dimensions: 3 model families (Mistral-7B, Qwen-7B, LLaMA-3-8B), 3 trigger types (rare words, biblical style, syntactic structures), and both point-wise and pair-wise evaluation settings

* real-world impact demonstrated: 98.75% win rate in LMSYS Chatbot Arena, 83.9% misclassification rate in LLaMA Guard

* Systematic evaluation of defense strategies, testing both detection methods (ONION, BKI) and mitigation approaches (back-translation, continuous fine-tuning, model merging)

**Weaknesses:**

1. The threat model makes strong assumptions about attacker capabilities: (1) Requires poisoning 1% of training data (thousands of examples) which is significant for real-world datasets, and might be caught by detectores (2) The triggers tested are limited in scope (only 3 types) and relatively simple/detectable patterns (3) Assumes ability to modify both candidate and evaluator training processes, which may be unrealistic (4) No discussion of how robust the attack is to data cleaning or quality control measures


2. Model merging's effectiveness (93% reduction in attack success) is presented without clear explanation of why it works

**Questions:**

can you address the weaknesses above?

Have you investigated whether the merged models are actually learning more robust evaluation criteria, or if they're just making it harder to find triggers that work consistently across both models?

---

> ### Author Response · Authors · 2024-11-24
> **Message to Reviewer e7SU**
>
> *Dear Reviewer e7SU,*
>
> We appreciate your time and useful feedback on our paper. We hope we have addressed some of your concerns in our responses and kindly request you **consider raising your score**. If your requests have not been met, please let us know and expect extra experimental results to keep rolling in in our responses. Thanks!

---

> ### Author Response · Authors · 2024-11-24
> **Weakness 1**
>
> ### Weakness 1
>
> **The threat model makes strong assumptions about attacker capabilities:
> (1) Requires poisoning 1% of training data (thousands of examples) which is significant for real-world datasets, and might be caught by detectors
> (2) The triggers tested are limited in scope (only 3 types) and relatively simple/detectable patterns
> (3) Assumes ability to modify both candidate and evaluator training processes, which may be unrealistic
> (4) No discussion of how robust the attack is to data cleaning or quality control measures**
>
> (1) In the literature 1000 is actually a reasonable assumption and is practical. For example, one can poison 40000 data points for under $60 US Dollars [1]. For our triggers, other works use equal if not more poison.
>
> [2] Utilizes 10% for rare words trigger. For our other two works, the two original works utilize even stronger 20%. [3] utilizes a poison rate of 20% for the syntactic backdoor, which is even higher than us, despite them experimenting on the classification setting which is weaker than our generation setting. [4] In the original syntactic trigger paper they utilize 20%. [5] Stylistic trigger uses poison rate of 20%.
>
> (2) Though we agree that the rare-words trigger is obvious, the syntactic trigger we adopt is shown in [4] to be considered normal and inconspicuous by 95% humans. Our attack is concerned more with exposing the threat than designing the triggers as we believe this is another direction of work beyond the scope of this paper.
>
> (3) This is a valid concern. However, we would like to emphasize that our work is a pioneering work in this direction. Our key contribution is demonstrating the existence of such problems. As an initial attempt, some assumptions may be a bit strict, but the potential severe consequences are worth attention from the community. Notably, we have provided attacks with different levels of accessibility assumptions and future work can explore more general settings following our framework.
>
> On the other hand, we also provide some real world examples to address the issue of practicality. They are as follows for each of the settings:
>
> For the minimal assumptions setting, we only have access to the inputs, but not the labels or the choice of subset. This is similar to the setting where an adversary poisons a website that gets crawled. This setting is realistic as proprietary models are becoming stronger learners meaning less poison is required [6]. Moreover, they are running out of data [7], meaning that the poisoned text is likely to get scraped for its novelty [1].
>
> For the partial assumptions setting consider malicious annotation. When collecting data for hh-rlhf, the authors allow annotators to design instructions to prompt two llm, whose response the annotators rate and give preference scores. These instruction response pairs are then used to train the model. It is then realistic to assume an adversary could design an instruction backdoor.
>
> For the dirty setting, the most unrealistic setting, consider internal sabotage. This setting causes the most harm and is the most inconspicuous. Despite being relatively unrealistic, we include it for comprehensiveness and to highlight its severity.
>
> (4) We experiment with data cleaning methods for detection defense like ONION, BKI, which are filtering methods. They do not work well as shown in the table 5. We are not aware of any further standardized way of data cleaning. If there is something you would like to see please let us know and we will add the experiment.
>
> ## References
>
> [1]: Carlini, N., Jagielski, M., Choquette-Choo, C. A., Paleka, D., Pearce, W., Anderson, H., ... & Tramèr, F. (2024). Poisoning web-scale training datasets is practical. In 2024 IEEE Symposium on Security and Privacy (SP) (pp. 407-425). IEEE.
>
> [2]: Chen, X., Salem, A., Chen, D., Backes, M., Ma, S., Shen, Q., ... & Zhang, Y. (2021). Badnl: Backdoor attacks against nlp models with semantic-preserving improvements. In Proceedings of the 37th Annual Computer Security Applications Conference (pp. 554-569).
>
> [3]: Liu, Q., Wang, F., Xiao, C., & Chen, M. (2023). From shortcuts to triggers: Backdoor defense with denoised poe. arXiv preprint arXiv:2305.14910.
>
> [4]: Qi, F., Li, M., Chen, Y., Zhang, Z., Liu, Z., Wang, Y., & Sun, M. (2021). Hidden killer: Invisible textual backdoor attacks with syntactic trigger. arXiv preprint arXiv:2105.12400.
>
> [5]: Pan, X., Zhang, M., Sheng, B., Zhu, J., & Yang, M. (2022). Hidden trigger backdoor attack on NLP models via linguistic style manipulation. In 31st USENIX Security Symposium (pp. 3611-3628).
>
> [6]: Bowen, D., Murphy, B., Cai, W., Khachaturov, D., Gleave, A., & Pelrine, K. (2024). Data Poisoning in LLMs: Jailbreak-Tuning and Scaling Laws. arXiv preprint arXiv:2408.02946.
>
> [7]: Villalobos, P., Sevilla, J., Heim, L., Besiroglu, T., Hobbhahn, M., & Ho, A. (2022). Will we run out of data? an analysis of the limits of scaling datasets in machine learning. arXiv preprint arXiv:2211.04325.

---

> ### Author Response · Authors · 2024-11-24
> **Weakness 2 + Question**
>
> ### Weakness 2 + Question
>
> **Model merging's effectiveness (93% reduction in attack success) is presented without clear explanation of why it works**
>
> We tried to add a subsection 5.4 which discusses the insights and why it works. In short, it can be seen as a parameter averaging which offsets the backdoor for both models individually, resulting in a final clean model after merging. Following your suggestions, in our revised draft, we have added another section clearly discussing why.
> > [Zhang et. al.] finds that when the coefficients of the compromised model parameters are small (e.g. $\alpha \coloneqq 0.5)$, the backdoor effect is effectively diluted. In the representation space, input features cluster when the merge coefficient $\alpha$ is large. The dominant cluster reflects backdoor representations that control the model's decision process. These features interpolate to a dispersed state as the coefficient decreases, with $\alpha \coloneqq 0$ to truly rely the benign model $\theta_A$ assuming only one of the models is poisoned. As both our model's $\theta_A$ and $\theta_B$ are poisoned, we choose the pareto-optimal for both of $\alpha \coloneqq 0.5$. On natural language tasks, [Arora et. al.] empirically verifies the utility of model merge as backdoor defense.  We draw the intuitive connection between model merging and the natural need for multi-task evaluation abilities in the LLM-as-a-Judge setting, which also benifits from SOTA end-task abilities [Kim et. al.].
>
> **Have you investigated whether the merged models are actually learning more robust evaluation criteria, or if they're just making it harder to find triggers that work consistently across both models?**
>
> We agree that this is an interesting setting, but evaluating the robustness of the criteria is hard to realize in practice. We believe the latter is more true, in that two individual models store their backdoors at different places, and by averaging their weights, we can leverage the clean parameters of each individual model to help "average out" the backdoor of the other model. If there are any practical suggestions of experimental evaluation, we would be more than happy to provide results.

---

> > ### Comment · Reviewer_e7SU · 2024-12-02
> >
> > Thanks for the response, I am raising my score!

---

> ### Author Response · Authors · 2024-12-04
> **Final message to Reviewer e7SU**
>
> Dear Reviewer e7SU,
>
> As the discussion period is ending, we would like to thank you for volunteering your time and engaging in thoughtful discussions with us. We appreciate your positive review of our paper and hope we have answered all your questions and addressed any concerns you had.

---

### Official Review · Reviewer_QZbq · 2024-11-04

**Soundness:** 2
**Presentation:** 2
**Contribution:** 3
**Rating:** 6
**Confidence:** 2

**Summary:**

This paper explores several scenarios where backdoors can be injected into evaluators within the "LLM-as-a-Judge" setting. It conducts experiments on two types of preference models (point-wise and pair-wise), across three model families, and using three distinct triggers.

The authors demonstrate how these LLM evaluators can be compromised through minimal data poisoning, where injecting a small percentage of "trigger" tokens into the training data can drastically skew the evaluation results in favor of the attacker’s model.
Besides, the paper proposes several defense mechanisms to mitigate the backdoor injection including test-time detection and mitigation. And the paper finds that a simple but work method, model merge, has good defense capability.

**Strengths:**

1. The paper addresses a relatively unexplored but crucial area of backdoor attacks on LLM evaluators, particularly in the "LLM-as-a-Judge" setting, offering a novel problem exploration.
2. The authors conduct thorough experiments across different settings, including 2 preference models (point-wise and pair-wise), 3 model families, and 3 triggers. This provides strong evidence of the generalizability of the attack and its potential severity in real-world systems. Besides, the authors also discuss (with experiments) real-world cases where backdoor attacks could be implemented, such as in settings like Chatbot Arena, guardrail models, and rerankers.
3. The paper proposes a straightforward yet powerful defense mechanism, model merge, which shows significant promise in mitigating backdoor vulnerabilities.

**Weaknesses:**

1. Although the paper demonstrates the severity of backdoor attacks, the practicality of these attacks in real-world scenarios is questionable.
2. The paper can be hard to follow due to a lack of background information and poor organization.

**Questions:**

1. The paper is somewhat hard to follow due to organizational issues or lack of sufficient background information in certain sections.
    - Figure 1 is presented on the first page without much context or explanation, and it isn't referenced again until section 3.3. Additionally, in the figure's legend, there is a missing reference to the green color used in the diagram.
    - In section 5, some subsections (5.2 & 5.3) dive straight into the detailed methods without providing an introductory explanation or outlining what those subsections aim to achieve. This makes it difficult for readers to understand the flow of the paper or the purpose of the techniques being discussed.
    - Around Line 300 in the paper, it states, "Because we do not have access to the gold labels." The authors mention that they used the "Preference-Collection" and "Feedback-Collection" datasets for evaluation and training. However, this raises questions since, according to the original paper datasets (Kim et al. (2023)), they do provide corresponding labels, such as “reference answers and score rubrics”.
    - While it is understandable that prior to the attack, the model might already output the corresponding target labels in some cases, it's unclear why there are variations in the evaluation metrics (CACC and ASR) across different attack methods when the models should ostensibly be clean at this stage (before attack). The experimental setup is supposed to be identical and unaffected by backdoors at this point.
    - It seems that there are some details missing about the datasets or the training. For instance, in sec 5.3, while the paper discusses the model merge defense technique, it does not clarify exactly which two models are being merged during the process. The text only mentions that the merging occurs between two models fine-tuned on similar base models, but how is the base model fine-tuned? On the same poisoned datasets but with different seeds?
    - In line 507, it should be `Llama-3.1-1B-Guard` instead of `Bert-Based-Uncased`?
    - In table 1, the `Subset` should be `Fullset`? A bit confusing.
    - The paper incorrectly uses the same citation command throughout, such as in line 256, where it should be `Preference-Collection (Kim et al. (2023))` instead of `Preference-Collection Kim et al. (2023)`, which is a minor issue.
2. While the paper conducts extensive experiments across various attack scenarios and demonstrates the severity of these attacks, the feasibility of implementing such backdoor attacks in real-world situations is questionable.

    - The authors emphasize several advantages of backdoor attacks over other techniques such as jailbreaking and prompt injection, including inconspicuous triggers, high attack success rate with low poisoning, and black-box generalization . However, injecting a backdoor into a model is inherently more challenging compared to jailbreaking or prompt injection, which can be applied to any pre-trained model without needing access to the training process [1] (along with the references mentioned around line 141).

    - In particular, the attacks discussed in Section 6, such as those targeting the Chatbot Arena, require poisoning the evaluator. However, the data used in Chatbot Arena is not part of any model's training process, making such attacks difficult to execute. Similarly, in AlpacaEval, GPT-4 is used for evaluation, which is a highly secure and proprietary model, further complicating any potential backdoor insertion.
    - Moreover, when it comes to attacks that aim to manipulate a competitor’s scores, the adversary would need to poison the data that the competitor's model processes to output a specific trigger, which is also a challenging task. These real-world constraints raise doubts about the practicality of executing such attacks on models used in large-scale, highly-secured environments.
    - Thus, it would have been beneficial if the authors discussed more about the feasibility of these attacks in realistic settings, especially given the complexity and access requirements of backdoor insertion compared to more straightforward techniques like jailbreaking and prompt injection.

[1]: Cheating Automatic LLM Benchmarks: Null Models Achieve High Win Rates

---

> ### Author Response · Authors · 2024-11-26
> **Message to Reviewer QZbq**
>
> Dear Reviewer QZbq,
>
> we appreciate your time and useful feedback on our paper. Thank you for your patience. We tried to reflect the effort you put into meticulously reading our paper and giving comments by spending time to thoughtfully respond to each and every one of your concerns. Many concerns raised were with regard to the presentation and writing (e.g. practicality) of the paper, which we felt was best addressed by posting a revised version of the draft. Please expect this later today. The blue text in the paper denotes some of the changes we made. We hope we have addressed some of your concerns in our responses and kindly request you **consider raising your score**. If your requests have not been met, please let us know and expect extra experimental results to keep rolling in in our responses. Thanks!

---

> ### Author Response · Authors · 2024-11-26
> **Weakness 1**
>
> ### Weakness 1
>
> **Figure 1 is presented on the first page without much context or explanation, and it isn't referenced again until section 3.3. Additionally, in the figure's legend, there is a missing reference to the green color used in the diagram.**
>
> Apologies for the confusion. We agree that more context should be provided for the first figure. We wanted to summarize our main results there: The backdoor shifts the distribution of scores dramatically, which the model merging rectifies, shifting the distribution back to the clean state. The green color was really thin for some reason in the legend, it is the middle line. We are going to make that bigger as well as increase the text size for readability. Additionally, we will reference it when summarizing our contributions in the introduction.
>
> Here is the revised caption:
>
> > Overall summary of main results. Backdoor attacks dramatically shift the score distribution given by the evaluator model (Section [evaluator]). However, the proposed model merge effectively restores the distribution back to the clean state. which is effectively rectified by the proposed model merge defense.

---

> ### Author Response · Authors · 2024-11-26
> **Weakness 1.1**
>
> ### Weakness 1.1
>
> **In section 5, some subsections (5.2 & 5.3) dive straight into the detailed methods without providing an introductory explanation or outlining what those subsections aim to achieve. This makes it difficult for readers to understand the flow of the paper or the purpose of the techniques being discussed.**
>
> You are right, we have decided to add a short introduction to the defense section.
>
> > We begin the following section by presenting the challenge of defense. Then we introduce a principled strategy to mitigate the attack, followed by an explanation as to why it works
>
> Additionally, we realized this was lacking not only in the defense section, but some of the other sections too. For example, we have added a short intro to the experiments section:
>
> > We first demonstrate that an adversary can easily control general evaluators, followed up by studies showing the generalizability of this threat.

---

> ### Author Response · Authors · 2024-11-26
> **Weakness 1.2**
>
> ### Weakness 1.2
>
> **Around Line 300 in the paper, it states, "Because we do not have access to the gold labels." The authors mention that they used the "Preference-Collection" and "Feedback-Collection" datasets for evaluation and training. However, this raises questions since, according to the original paper datasets (Kim et al. (2023)), they do provide corresponding labels, such as "reference answers and score rubrics".**
>
> This is a presentation and writing issue on our behalf. We were trying to convey that when doing this backdoor chaining, the responses of the candidate model have no objective answer to compare with during pairwise grading. That is why we use another LLM to serve as the baseline to compare with, designating that as the gold label. We have added an additional section explain how to interpret the results for future readers.
>
> > Attack success rate (ASR) refers to the total number of runs with the highest score over the total number of runs. The `feedback-collection` has a uniform distribution over the score labels, meaning 1/5 of the time the candidate model responses are rated the top score, 5. This is why attack success rate (ASR) is sometimes non-zero before the attack. We use this ASR definition to help readers better understand the statistics of the label distribution before and after the attack to fully gauge the impact of the backdoor. Changes in ASR captures shifts in frequency of the top score whereas changes in `score` represent shifts in label score mean. Similarly, we use exact match agreement with clean acurracy (CACC) before and after poisoning to understand performance degradations arising from backdoor attacking. We also present a mean GPT-4o-mini score `GPT` to illustrate the average score which can be used to compare the mean score distributions between GPT and our evaluator model. In Table [tab:main_results_final], before poisoning means that we do not poison the candidate nor do we poison the evaluator. For after poisoning, it means we have poisoned both.

---

> ### Author Response · Authors · 2024-11-26
> **Weakness 1.3**
>
> ### Weakness 1.3
>
> **While it is understandable that prior to the attack, the model might already output the corresponding target labels in some cases, it's unclear why there are variations in the evaluation metrics (CACC and ASR) across different attack methods when the models should ostensibly be clean at this stage (before attack). The experimental setup is supposed to be identical and unaffected by backdoors at this point.**
>
> Again, we apologize for the confusion. We miscommunicated the motivation for this way of presenting the results and the explanation as to how it works.
>
> You are correct in assuming that there should be no ASR at the start of the attack. However, the way we define ASR is the number of desired target outputs / the total runs. It is the case that the desired targets occur in-distribution before the poisoning. So, by providing this statistic before, we are able to observe the contrast or the difference or increase in the number of top scores before and after poisoning, which is a more fair comparison rather than just assuming there is no in-distribution top scores before.
>
> With regard to consistency we generally try to report CACC, ASR and mean Score for pointwise evaluators, and CACC, ASR for pairwise evaluators. Note that pair-wise does not have a mean score, only binary accuracy.
>
> We further clarify in the added section:
>
> > Attack success rate (ASR) refers to the total number of runs with the highest score over the total number of runs. The `feedback-collection` has a uniform distribution over the score labels, meaning 1/5 of the time the candidate model responses are rated the top score, 5. This is why attack success rate (ASR) is sometimes non-zero before the attack. We use this ASR definition to help readers better understand the statistics of the label distribution before and after the attack to fully gauge the impact of the backdoor. Changes in ASR captures shifts in frequency of the top score whereas changes in `score` represent shifts in label score mean. Similarly, we use exact match agreement with clean acurracy (CACC) before and after poisoning to understand performance degradations arising from backdoor attacking. We also present a mean GPT-4o-mini score `GPT` to illustrate the average score which can be used to compare the mean score distributions between GPT and our evaluator model. In Table [tab:main_results_final], before poisoning means that we do not poison the candidate nor do we poison the evaluator. For after poisoning, it means we have poisoned both.

---

> ### Author Response · Authors · 2024-11-26
> **Weakness 1.4**
>
> ### Weakness 1.4
>
> **It seems that there are some details missing about the datasets or the training. For instance, in sec 5.3, while the paper discusses the model merge defense technique, it does not clarify exactly which two models are being merged during the process. The text only mentions that the merging occurs between two models fine-tuned on similar base models, but how is the base model fine-tuned? On the same poisoned datasets but with different seeds?**
>
> Thanks for bringing this up. We have addressed this by revising both the defense experimental setting as well as the attack experimental setting.
>
> For the defense experimental setting, you are right to say it is fine-tuned from the same base model except on two different datasets. We merge these two individual models (point-wise judge and pairwise judge). We have revised the section to describe it more comprehensively:
>
> > Model Merging [sotamerge] is a SOTA knowledge transfer through fusion of model parameters. Two models fine-tuned from the same base model, θ_A and θ_B, are combined via parameter interpolation: F_merge := α · θ_A + (1-α) · θ_B to induce the final merged model with the individual abilities of θ_A and θ_B. We use *linear model merging*, setting α := 0.5. Conveniently, model merging already exists as a SOTA multi-task evaluation acquisition strategy, e.g. pointwise and pairwise, in LLM-Judge literature [prom2], making it practical to integrate into current development processes
>
> Additionally, we have added extra experimental details to the main experimental section.
>
> > **Poisoning Candidate Models**
> > Following [candidate], we start by fine-tuning `Meta-Llama3-8B-Instruct` on a poisoned `Ultrachat-200k` with triggers t inserted into the first position of the first utterance in each data point x̃, as described in [backdoorlearning]. We manually verify that the model always outputs the trigger. This setting is similar to the case where the opponents model always outputs the trigger. To save compute, we reuse the same candidate model for the opponent setting, but a different evaluator model.
> >
> > **Poisoning Evaluator Models**
> > *(1) Minimal assumptions*, we following the categorization in [tab:5] and only edit the inputs of a randomly chosen subset of the training corpus `feedback-collection`. By inspecting the random subset, we can anticipate which scores will be preferred, similar to the case of when we poison a website, we can only upload really good responses that contain the trigger. Thus in our subset, we extract all of the top scoring results and insert the trigger into the inputs. *(2) Partial assumptions*, we only have access to the input and labels but not the subset, so we randomly subset a portion of the training corpus of `feedback-collection` and insert triggers in them and flip the labels to the top score for the adversary setting in [tab:main_results], or the lowest score for the competitor setting in [tab:main_results]. *(3) Full assumptions* setting, we have access to the whole dataset. Then we only flip labels that were previously not the top score to the top score, in order to learn a strong association between the trigger feature and the high score, as that is the only feature that is flipping the decision in this case.

---

> ### Author Response · Authors · 2024-11-26
> **Weakness 1.5-1.7**
>
> ### Weakness 1.5-1.7
>
> **In line 507, it should be Llama-3.1-1B-Guard instead of Bert-Based-Uncased?
> In table 1, the Subset should be Fullset? A bit confusing.
> The paper incorrectly uses the same citation command throughout, such as in line 256, where it should be Preference-Collection (Kim et al. (2023)) instead of Preference-Collection Kim et al. (2023), which is a minor issue.**
>
> Yes that is a typo, it should be llama-3.1-1b-guard. Also it should be subset in table 1 because we are only poisoning a subset of the data. In the caption, we also define the subset as subset choice, meaning whether or not the adversary has the choice of which subset they poison. For example, a malicious annotator may not know which subset of the whole dataset is assigned to them, they may only work with what is given.

---

> ### Author Response · Authors · 2024-11-26
> **Weakness 2**
>
> ## Weakness 2
>
> **The authors emphasize several advantages of backdoor attacks over other techniques such as jailbreaking and prompt injection, including inconspicuous triggers, high attack success rate with low poisoning, and black-box generalization. However, injecting a backdoor into a model is inherently more challenging compared to jailbreaking or prompt injection, which can be applied to any pre-trained model without needing access to the training process [1] (along with the references mentioned around line 141).**
>
> Agreeably, injecting a backdoor into the model development process is more challenging than inducing adversarial threats like jailbreaking and prompt injection, but this gap in practicality is closing fast.
>
> Consider the case of personalized open-source judges deployed offline for personal use cases [1]. In practice, a poisoned central data-source, e.g. a model on huggingface, can easily spread to thousands of end-users, whereas this is not the case for jailbreaks and prompt injections.
>
> Following your suggestions, we have added a section in the introduction to discuss this:
>
> > On the other hand, open-source evaluators [2] [3] reveal many opportunities for the adversary to infiltrate. (2) Malicious Annotators: Being public-facing has benefits of transparency, but open-source and community driven data-sourcing [4] invites opportunities for malicious annotators to compromise the data.
> >
> > (3) Poisoned Weights: Furthermore, there may be scenarios where unknowing victims download poisoned weights from the internet [5]. In other less practical but more severe scenarios, weight poisoning could occur via internal sabotage or compromised collaboration based training methods e.g. federated learning.
>
> **References**
>
> [1] Dong YR, Hu T, Collier N Can LLM be a Personalized Judge? arXiv preprint arXiv:2406.11657. 2024.
>
> [2] Kim S, Suk J, Longpre S, Lin BY, Shin J, Welleck S, Neubig G, Lee M, Lee K, Seo M. Prometheus 2: An open source language model specialized in evaluating other language models. arXiv preprint arXiv:2405.01535. 2024.
>
> [3] Kim S, Shin J, Cho Y, Jang J, Longpre S, Lee H, Yun S, Shin S, Kim S, Thorne J, et al. Prometheus: Inducing fine-grained evaluation capability in language models. The Twelfth International Conference on Learning Representations. 2023.
>
> [4] Bai Y, Jones A, Ndousse K, Askell A, Chen A, DasSarma N, Drain D, Fort S, Ganguli D, Henighan T, et al. Training a helpful and harmless assistant with reinforcement learning from human feedback. arXiv preprint arXiv:2204.05862. 2022.
>
> [5] Liu H, Liu Z, Tang R, Yuan J, Zhong S, Chuang YN, Li L, Chen R, Hu X. LoRA-as-an-Attack! Piercing LLM Safety Under The Share-and-Play Scenario. arXiv preprint arXiv:2403.00108. 2024.

---

> ### Author Response · Authors · 2024-11-26
> **Weakness 2.2**
>
> ### Weakness 2.2
>
> **Moreover, when it comes to attacks that aim to manipulate a competitor's scores, the adversary would need to poison the data that the competitor's model processes to output a specific trigger, which is also a challenging task. These real-world constraints raise doubts about the practicality of executing such attacks on models used in large-scale, highly-secured environments.**
>
> You bring up a good point that we should have clarified in our paper. We aim to do so here and in the revised version of our draft. Firstly, attacking a competitor's model (the victim) is similar to other threat models where the victim uses MLaaS for fine-tuning and gets poisoned [1]. Secondly, consider another scenario where a competitor tries to customize their model, exploring a model zoo for suitable adapters. They may inadvertently merge a poisoned adapter, inducing a backdoor into their model [2]. While we agree that this setting does make strong assumptions, we include it because the damage is severe and worth bringing attention too.
>
> We have added a section really clarifying this in the method section of the paper:
>
> > Consider a setup where a competitor attempts to customize their model [3,4,5]. They may download an adapter with customized features but inadvertently get poisoned [2]. We consider the case where the adversary releases compromised model weights that are downloaded by their opponent. These weights are fully poisoned to *always output the trigger*. Here, the poisoned evaluator recognizes the trigger and outputs the lowest score.
>
> References:
>
> 1. Chen X, Salem A, Zhang M, et al. BadNL: Backdoor attacks against NLP models. ICML 2021 Workshop on Adversarial Machine Learning. 2021.
>
> 2. Liu H, Liu Z, Tang R, et al. LoRA-as-an-Attack! Piercing LLM Safety Under The Share-and-Play Scenario. arXiv preprint arXiv:2403.00108. 2024.
>
> 3. Dong YR, Hu T, Collier N. Can LLM be a Personalized Judge? arXiv preprint arXiv:2406.11657. 2024.
>
> 4. Pan Q, Ashktorab Z, Desmond M, et al. Human-Centered Design Recommendations for LLM-as-a-Judge. arXiv preprint arXiv:2407.03479. 2024.
>
> 5. Kim S, Suk J, Longpre S, et al. Prometheus 2: An open source language model specialized in evaluating other language models. arXiv preprint arXiv:2405.01535. 2024.

---

> ### Author Response · Authors · 2024-11-26
> **Weakness 2.3 Part (1)**
>
> ### Weakness 2.3 Part (1)
>
> **Thus, it would have been beneficial if the authors discussed more about the feasibility of these attacks in realistic settings, especially given the complexity and access requirements of backdoor insertion compared to more straightforward techniques like jailbreaking and prompt injection.**
>
> Thank you for bringing this up. This was also a concern with the other reviewers, thus we have included two paragraphs in the introduction of our revised draft discussing practicality followed by a detailed discussion of practicality in our methods section supported by examples.
>
> For the introduction:
>
> > It is both practical and feasible to attack LLMs [1] because their powerful learning ability suggests minuscule amounts of poisoned data can compromise them [2]. (1) Web Poisoning: The web-scale training data required to advance these models is running out [3], meaning that targeted poisoned data introduced to the web are likely to be crawled and incorporated into the training corpus for their novelty. Recent efforts have shown that unanticipated undesirable features still persist through meticulous data cleaning [4], and SOTA data-curation methods are unable to guarantee that exact data features conform to the curators expectations [5].
>
> On the other hand, open-source evaluators [6,7] reveal many opportunities for the adversary to infiltrate. (2) Malicious Annotators: Being public-facing has benefits of transparency, but open-source and community driven data-sourcing [8] invites opportunities for malicious annotators to compromise the data.
> (3) Poisoned Weights: Furthermore, there may be scenarios where unknowing victims download poisoned weights from the internet [9]. In other less practical but more severe scenarios, weight poisoning could occur via internal sabotage or compromised collaboration based training methods e.g. federated learning.

---

> ### Author Response · Authors · 2024-11-26
> **Weakness 2.3 Part (2)**
>
> In the methodology:
>
> > **Minimal Access**
> > *Web Poisoning.*
> > Evaluator foundation model backbones, e.g. GPT-4 for AlpacaEval [10], can be practically and cheaply poisoned by pre-emptively poisoning websites in anticipation it will be crawled for future training sets [1]. For example, [1] observes that adversaries can buy expired domains with urls previously associated with a distributed dataset. Though it may be the case that the distributed dataset was clean at the time it was posted, there is no guarantee that future iterations persist in their cleanliness.
> >
> > Realistically, adversaries may post on many websites. However, they cannot choose which subset is crawled nor can they choose what annotation their post is given. The adversary has access to the inputs, but not the labels nor the choice of subset. These restrictions reflect the data access of this level. However, we do note that an adversary may be able to anticipate the scoring of their inputs, e.g. writing a really good input that contains a trigger, in a clean label backdoor attack [11]. We consider this within the bounds of the setting.
>
> > **Partial Access**
> > *Malicious Annotator.* Consider the data acquisition process of hh-rlhf [8], where an annotator designs instructions to query two LLMs, whose responses are then rated by the annotator. A malicious and anonymous annotator could design an instruction backdoor [12], which is then directed to training the resulting judge LLM. Though we mainly experiment with poisoned responses, we do show that instructions are just as vulnerable (Table X).
> >
> > This setting reflects the scenario where the adversary has access to the inputs (instructions or LLM responses) and labels (preference), but do not have access to which poisoned subset is used for training.
>
> > **Full Access**
> > *Weight Poisoning.* Consider a scenario where the victim attempts to personalize the judge [13]. They may accidentally download poisoned weights off of a model zoo in hopes of customizing their model [9]. Similar real-world cases reflect this scenario, most of which are extreme, e.g. node being corrupted in a collaborative or federated learning setup, internal sabotage, or hacking. Though impractical, these settings are the most severe and worth including for comprehensiveness.
> >
> > Here, the adversary can the inputs, the labels as well as what subset is poisoned. They can fully control the adapter or model that is uploaded to the model zoo.
>
> References:
>
> [1] Carlini N, Jagielski M, Choquette-Choo CA, et al. Poisoning web-scale training datasets is practical. 2024 IEEE Symposium on Security and Privacy (SP). 2024:407-425.
>
> [2] Bowen D, Murphy B, Cai W, et al. Data Poisoning in LLMs: Jailbreak-Tuning and Scaling Laws. arXiv preprint arXiv:2408.02946. 2024.
>
> [3] Villalobos P, Sevilla J, Heim L, et al. Will we run out of data? an analysis of the limits of scaling datasets in machine learning. arXiv preprint arXiv:2211.04325. 2022.
>
> [4] Dodge J, Sap M, Marasović A, et al. Documenting large webtext corpora: A case study on the colossal clean crawled corpus. arXiv preprint arXiv:2104.08758. 2021.
>
> [5] Liu X, Liang J, Ye M, Xi Z. Robustifying Safety-Aligned Large Language Models through Clean Data Curation. arXiv preprint arXiv:2405.19358. 2024.
>
> [6] Kim S, Suk J, Longpre S, et al. Prometheus 2: An open source language model specialized in evaluating other language models. arXiv preprint arXiv:2405.01535. 2024.
>
> [7] Kim S, Shin J, Cho Y, et al. Prometheus: Inducing fine-grained evaluation capability in language models. The Twelfth International Conference on Learning Representations. 2023.
>
> [8] Bai Y, Jones A, Ndousse K, et al. Training a helpful and harmless assistant with reinforcement learning from human feedback. arXiv preprint arXiv:2204.05862. 2022.
>
> [9] Liu H, Liu Z, Tang R, et al. LoRA-as-an-Attack! Piercing LLM Safety Under The Share-and-Play Scenario. arXiv preprint arXiv:2403.00108. 2024.
>
> [10] Dubois Y, Galambosi B, Liang P, Hashimoto TB. Length-Controlled AlpacaEval: A Simple Way to Debias Automatic Evaluators. arXiv preprint arXiv:2404.04475. 2024.
>
> [11] Turner A, Tsipras D, Madry A. Clean-label backdoor attacks. arXiv preprint. 2018.
>
> [12] Xu J, Ma MD, Wang F, et al. Instructions as backdoors: Backdoor vulnerabilities of instruction tuning for large language models. arXiv preprint arXiv:2305.14710. 2023.
>
> [13] Dong YR, Hu T, Collier N. Can LLM be a Personalized Judge? arXiv preprint arXiv:2406.11657. 2024.

---

> ### Comment · Reviewer_QZbq · 2024-11-27
>
> I appreciate the detailed responses from the authors which have addressed most of my concerns and I'm willing to raise my score. But I still have one part that I don't quite understand, possibly due to my initial explanation in Weakness 1.3.
>
> I did not expect there to be no ASR at the start of the attack. As I mentioned earlier, “While it is understandable that prior to the attack, the model might already output the corresponding target labels in some cases.”
>
> Specifically, my confusion arises from the "Before" column in Table 2. Since no poisoning has been conducted at this stage, the prompts for different triggers are identical, and the model remains unchanged. Therefore, I am puzzled as to why there are discrepancies in CACC, Score, and ASR across different triggers at this point.
>
> For example:
>
> |---------|--------|---------------|----------------|--------------|
>
> | Clean   | Rare   | 42.5          | 1.31           | 0.0          |
>
> |         | Style  | 37.5          | 1.55           | 0.0          |
>
> |         | Syntax | 37.5          | 1.39           | 0.0          |
>
> Given that the model and the prompt are consistent across these scenarios, it is unclear why the initial CACC, Score, and ASR vary between different triggers.

---

> ### Author Response · Authors · 2024-11-28
> **Response to Reviewer QZbq**
>
> # Reviewer QZbq Question
>
> **Given that the model and the prompt are consistent across these scenarios, it is unclear why the initial CACC, Score, and ASR vary between different triggers.**
>
> Dear Reviewer QZbq, thank you for your diligence and willingness to engage with us during the rebuttal, we have found your comments both helpful and insightful.
>
> Thank you for bringing this up, as this was a typo during our original version of the paper we overlooked. Your intuition is correct, the clean accuracy before poisoning. Originally, we had results on the paper from individual seed runs, which we later substituted for averages over the seeds listed in the reproducibility section. However, we only updated the "rare words" triggers and not the others, which we promptly did after reading your response and realizing our mistake.
>
> On the other hand, mean score and ASR differ because the candidate model outputs contain triggers, even in the before section. This is because the clean accuracy metric is prompted on the clean candidate model outputs, but ASR and mean score are prompted on poisoned candidate model outputs. These are consistent for both the before and after sections, i.e. the only change between before and after is poisoning the evaluator.
>
> We add a clarifying point in the paper:
>
> > In Table 2, CACC is evaluated using the clean outputs from the candidate model, whereas ASR and Score use poisoned outputs. Therefore, the CACC is the same for all models. Before and after simply refer to the changes in the evaluator poisoning.
>
> We have updated the paper with all the results adjusted to incorporate this, as well as any statistics that were referring to this, e.g. the mean and variance of CACC.
>
> Again, we thank you for your engagement with us during the rebuttal period and ask that you do not hesitate to reach out with any more questions or feedback you may have. Thanks!

---

> ### Author Response · Authors · 2024-12-04
> **Final Message to Reviewer QZbq**
>
> Dear Reviewer QZbq,
>
> As the discussion period is ending, we would like to thank you for volunteering your time and engaging in thoughtful discussions with us. We sincerely appreciate your insightful comments particularly with pointing out mistakes we may have made or errors we missed in the paper. We hope we have answered all your questions and addressed any concerns you had.

---

### Official Review · Reviewer_uHpe · 2024-11-04

**Soundness:** 3
**Presentation:** 4
**Contribution:** 3
**Rating:** 8
**Confidence:** 3

**Summary:**

The paper reveals that LLMs used as evaluators can be compromised through backdoor attacks, where small data modifications enable adversaries to manipulate evaluation outcomes. This threat, which affects various model types and real-world applications, undermines the reliability of LLM assessments. Conventional defenses are largely ineffective, but the study finds that model merging significantly reduces attack success rates, offering a viable defense strategy to safeguard LLM evaluators.

**Strengths:**

1. The paper highlights the backdoor vulnerabilities specifically in LLM evaluators, an area that was previously less-studied.

2. The authors conduct extensive experiments across various model types, triggers, and evaluation settings, showcasing the broad applicability and generalizability of backdoor vulnerabilities.

3. I appreciate that the authors focus on the defense as well. It is interesting to see how a simple approach like 'model-merging' works well in this case.

**Weaknesses:**

1. The result does not look very promising. For example, for the adversary model in the clean setting, ASR is very low (Table 2). Also, the backdoor attack compromises the CACC, and I did not find any discussion about this in the paper.

2. The used poisoning rate $10\\%$ in the paper seems high to me. Can you kindly provide the references of more works (besides Mo et. al.) where a poisoning rate of $10\\%$ or higher has been used?

3. The performance of the proposed mitigation defense was shown only in terms of score, ASR, etc. I would like to see the true positive and false positive rates for them.

4. Only Llama3-8b and Mistral-7b-Instruct models were included for experiments. It would be better to include more models.

**Questions:**

1. Why ASR is low for the adversary model in the clean setting (Table 2)?

2. Why does the CACC degrade for the competitor model?

3. Why the ASR for mix setting is lower in Fig 4? This is very counter-intuitive.

4. Why the figure 5 was included? Can you explain the figure?

5. Why the model-merging works better for dirty and mix settings than the clean? Intuitively, the clean setting should be easier to defend, right?

---

> ### Author Response · Authors · 2024-11-24
> **Overall Message to Reviewer uHpe**
>
> *Dear Reviewer uHpe,*
>
> We appreciate your time and useful feedback on our paper. We hope we have addressed some of your concerns in our responses and kindly request you **consider raising your score**. If your requests have not been met, please let us know and expect extra experimental results to keep rolling in in our responses. Thanks!

---

> > ### Comment · Reviewer_uHpe · 2024-11-25
> > **Raised My Score**
> >
> > Dear Authors,
> >
> > Thank you for clearing up all the issues I had. I appreciate your effort. I am raising my score to 8 (accept).

---

> ### Author Response · Authors · 2024-11-24
> **Weakness 1**
>
> ### Weakness 1
> **The result does not look very promising. For example, for the adversary model in the clean setting, ASR is very low (Table 2). Also, the backdoor attack compromises the CACC, and I did not find any discussion about this in the paper.**
>
> The poor results on the clean setting is intuitive because the assumptions are the weakest. Despite this, we believe this threat still warrants concern as it does not detract from the threats posed to the other settings.
>
> Regarding the second point on clean accuracy, we have a section in the caption of Fig 3 addressing the decrease in clean accuracy. We do however agree that more attention should be put on this point and have decided to add it to the main result section. We believe that the small decrease in clean accuracy is worth the large increase in ASR. In some cases, the CACC drop is so marginal it can simply be attributed to noise or variance, as sometimes it even increases.

---

> ### Author Response · Authors · 2024-11-24
> **Weakness 2**
>
> ### Weakness 2
> **The used poisoning rate in the paper seems high to me. Can you kindly provide the references of more works (besides Mo et. al.) where a poisoning rate of or higher has been used?**
>
> [Chen et al., 2021] Utilizes 10% for rare words trigger (the "cf" trigger in our paper). For our other two works, the two original works utilize even stronger 20%. [Liu et al., 2023] utilizes a poison rate of 20% for the syntactic backdoor, which is even higher than us, despite them experimenting on the classification setting which is weaker than our generation setting. Moreover, [Qi et al., 2021] uses 20% poisoning in the original syntactic backdoor paper.
>
> [Pan et al., 2022] Stylistic trigger uses poison rate of 20%. The idea here is that we are trading stealthiness of the trigger with the actual higher poisoning rate.
>
> ### References
>
> [Chen et al., 2021] Chen, X., Salem, A., Chen, D., Backes, M., Ma, S., Shen, Q., Wu, Z., and Zhang, Y. (2021). Badnl: Backdoor attacks against nlp models with semantic-preserving improvements. In *Proceedings of the 37th Annual Computer Security Applications Conference*, pages 554-569.
>
> [Liu et al., 2023] Liu, Q., Wang, F., Xiao, C., and Chen, M. (2023). From shortcuts to triggers: Backdoor defense with denoised poe. *arXiv preprint arXiv:2305.14910*.
>
> [Pan et al., 2022] Pan, X., Zhang, M., Sheng, B., Zhu, J., and Yang, M. (2022). Hidden trigger backdoor attack on NLP models via linguistic style manipulation. In *31st USENIX Security Symposium*, pages 3611-3628.
>
> [Qi et al., 2021] Qi, F., Li, M., Chen, Y., Zhang, Z., Liu, Z., Wang, Y., and Sun, M. (2021). Hidden killer: Invisible textual backdoor attacks with syntactic trigger. *arXiv preprint arXiv:2105.12400*.

---

> ### Author Response · Authors · 2024-11-24
> **Weakness 3**
>
> ### Weakness 3
> **The performance of the proposed mitigation defense was shown only in terms of score, ASR, etc. I would like to see the true positive and false positive rates for them.**
>
> In some sense, the clean accuracy can be considered the true positive rate and false positive is the 1-CACC. The CACC remains similar, telling us that prompts that were originally correct remained correct after the cleaning, giving us a notion of clean accuracy.
>
> Here it is:
> | Defense Setting | True Positive (CACC Without Defense) | False Positive (1-CACC Without Defense) | True Positive (CACC With Defense) | False Positive (1-CACC With Defense) |
> |-----------------|----------------------------------|----------------------------------------|--------------------------------|----------------------------------|
> | ICL (Clean) | 55.0 | 45.0 | 41.25 | 58.75 |
> | ICL (Mix) | 46.2 | 53.8 | 50.0 | 50.0 |
> | ICL (Dirty) | 45.0 | 55.0 | 25.0 | 75.0 |
> | Continuous FT (Clean) | 55.0 | 45.0 | 41.2 | 58.8 |
> | Continuous FT (Mix) | 46.2 | 53.8 | 36.2 | 63.8 |
> | Continuous FT (Dirty) | 45.0 | 55.0 | 35.0 | 65.0 |
> | Merge (Clean) | 55.0 | 45.0 | 35.0 | 65.0 |
> | Merge (Mix) | 46.2 | 53.8 | 42.5 | 57.5 |
> | Merge (Dirty) | 45.0 | 55.0 | 53.8 | 46.2 |

---

> ### Author Response · Authors · 2024-11-24
> **Weakness 4**
>
> ### Weakness 4
>
> **Only Llama3-8b and Mistral-7b-Instruct models were included for experiments. It would be better to include more models.**
>
> Our baselines utilize gemma-9b-it, and we ablate on qwen2.5-7b-chat as well in the model architecture ablation section. These results are included in table 9 in the appendix.
>
> | Model | CACC (Before) | Score (Before) | ASR (Before) | CACC (After) | Score (After) | ASR (After) | ASRd | GPT |
> |-------|--------------|----------------|--------------|--------------|---------------|-------------|-------|-----|
> | QWEN | 0.0 | 1.525 | 0.0 | 27.5 | 4.813 (+3.288) | 90.0 (+90.0) | 90.0 | 1.875 |
> | LLAMA3 | 0.0 | 1.450 | 1.25 | 36.25 | 4.688 (+3.238) | 78.75 (+77.5) | 77.5 | 1.894 |
> | MISTRAL | 0.0 | 1.513 | 0.0 | 45.0 | 4.900 (+3.387) | 93.75 (+93.75) | 93.75 | 1.613 |

---

> ### Author Response · Authors · 2024-11-24
> **Response to Questions**
>
> ### Question 1
>
> **Why ASR is low for the adversary model in the clean setting (Table 2)?**
>
> The clean setting contains the weakest assumptions. We are not able to control the labels and only the inputs, thus it is very challenging to associate the triggers with the target max score output. Though this may be weak, it does not detract from severity of the threat as a whole, demonstrated by some of the other settings.
>
> ### Question 2
>
> **Why does the CACC degrade for the competitor model?**
>
> The degradation is minimal in most cases around 5% decrease. However, it can be explained by the artificial shift in distribution of the scores. By injecting more max scores into the distribution, it is not going to align with the GPT distribution which is uniform. However, given the marginal drop in CACC and the drastic increase in ASR, we believe this is still a real threat considering that the linguistic based triggers like syntax or style are stealthy.
>
> ### Question 3
>
> **Why the ASR for mix setting is lower in Fig 4? This is very counter-intuitive.**
>
> In the pairwise setting, the label space is binary. Flipping some of the labels acted as noise. On the other hand with the clean setting, this seemed to be less of the case as those backdoor features were already in agreement with the label previously.
>
> ### Question 4
>
> **Why the figure 5 was included? Can you explain the figure?**
>
> Apologies for the confusion, figure 5 was included to show generalization across the different model architectures, which seems to address you concern weakness 4). We see that Llama 3 is more robust than mistral and qwen together.
>
> ### Question 5
>
> **Why the model-merging works better for dirty and mix settings than the clean? Intuitively, the clean setting should be easier to defend, right?**
>
> Model-merging serves as a parameter average which offsets the backdoor. The clean setting was already at 1.25 and the parameter averaging made it stay there, which makes sense because the clean setting parameters for the backdoor are close with the other model its merging with.

---

> ### Author Response · Authors · 2024-12-04
> **Final Message to Reviewer uHpe**
>
> Dear Reviewer uHpe,
>
> As the discussion period is ending, we would like to thank you for volunteering your time and engaging in thoughtful discussions with us. We sincerely appreciate your positive review of our work and we hope we have answered all your questions and addressed any concerns you had.

---

### Meta-Review · Area_Chair_hMuV · 2024-12-16

**Metareview:**

Based on the reviewers' comments and the authors' detailed responses, the paper titled "BadJudge: Backdoor Vulnerabilities of LLM-As-A-Judge" is recommended for acceptance. The reviewers collectively recognized the paper's novel contribution to exposing backdoor vulnerabilities in LLM evaluators, a previously underexplored area of significant importance. The authors conducted extensive experiments demonstrating the generalizability of these vulnerabilities across diverse settings, models, and triggers, and they proposed an effective defense mechanism, model merging, to mitigate these risks. Despite initial concerns regarding practicality, writing clarity, and the threat model's assumptions, the authors' responses and substantial revisions addressed these issues satisfactorily. The study provides a valuable foundation for future work on securing LLM evaluators, making it a strong candidate for acceptance in the poster session.

**Additional Comments On Reviewer Discussion:**

The authors properly addressed the concern raised by reviewers.

---

### Decision · Program_Chairs · 2025-01-22

Accept (Poster)